# A Novel Approach to Develop New and Potent Inhibitors for the Simultaneous Inhibition of Protease and Helicase Activities of HCV NS3/4A Protease: A Computational Approach

**DOI:** 10.3390/molecules28031300

**Published:** 2023-01-29

**Authors:** Muhammad Riaz, Ashfaq Ur Rehman, Muhammad Waqas, Asaad Khalid, Ashraf N. Abdalla, Arif Mahmood, Junjian Hu, Abdul Wadood

**Affiliations:** 1Computational Medicinal Chemistry Laboratory, Department of Biochemistry, Abdul Wali Khan University Mardan, Mardan 23200, Pakistan; 2School of Biological Science, University of California, Irvine, CA 92697, USA; 3Natural and Medical Sciences Research Center, University of Nizwa, Birkat Al Mauz, P.O. Box 33, Nizwa 616, Oman; 4Substance Abuse and Toxicology Research Center, Jazan University, P.O. Box 114, Jazan 45142, Saudi Arabia; 5Medicinal and Aromatic Plants and Traditional Medicine Research Institute, National Center for Research, Khartoum P.O. Box 2404, Sudan; 6Department of Pharmacology and Toxicology, College of Pharmacy, Umm Al-Qura University, Makkah 21955, Saudi Arabia; 7Center for Medical Genetics and Hunan Key Laboratory of Medical Genetics, School of Life Sciences, Central South University, Changsha 410078, China; 8Department of Central Laboratory, SSL, Central Hospital of Dongguan City, Affiliated Dongguan Shilong People’s Hospital of Southern Medical University, Dongguan 523000, China

**Keywords:** HCV NS3/4A protease, molecular docking, RECAP analyses, RECAP synthesis, inhibitors

## Abstract

Infection of hepatitis C (HCV) is a major threat to human health throughout the world. The current therapy program suffers from restricted efficiency and low tolerance, and there is serious demand frr novel medication. NS3/4A protease is observed to be very effective target for the treatment of HCV. A data set of the already reported HCV NS3/4A protease inhibitors was first docked into the NS3/4A protease (PDB ID: 4A92A) active sites of both protease and helicase sites for calculating the docking score, binding affinity, binding mode, and solvation energy. Then the data set of these reported inhibitors was used in a computer-based program “RECAP Analyses” implemented in MOE to fragment every molecule in the subset according to simple retrosynthetic analysis rules. The RECAP analysis fragments were then used in another computer-based program “RECAP Synthesis” to randomly recombine and generate synthetically reasonable novel chemical structures. The novel chemical structures thus produced were then docked against HCV NS3/4A. After a thorough validation of all undertaken steps, based on Lipinski’s rule of five, docking score, binding affinity, solvation energy, and Van der Waal’s interactions with HCV NS3/4A, 12 novel chemical structures were identified as inhibitors of HCV NS3/4A. The novel structures thus designed are hoped to play a key role in the development of new effective inhibitors of HCV.

## 1. Introduction

HCV infection is a serious health risk throughout the world that victimizes more than 200 million people [1,2,3]. It was observed in 1989 that Hepatitis C virus is a major causative mediator of hepatitis C [4]. Hepatitis C virus belongs to the flaviviridae family and is a positive-stranded RNA virus [5]. Hepatitis C Virus can live for decades within a host without any hazardous symptoms [6]. HCV infection was observed in about 10 million individuals in Pakistan [7]. The infection, eventually leads to serious complications like cirrhosis, liver failure, fibrosis in 60–70% of affected people [8]. Before the development of combination therapy of HCV protease inhibitor, patients with HCV infection were cured with pegylated interferon-α and ribavirin [9]. However, in a number of patients, the adverse side effects of this type of treatment may lead to its discontinuation [10]. Furthermore, it was found that only 50% of patients with genotype 1 infection successfully respond to this type of treatment [11,12]. Thus the adverse side effects and low competency of this type of treatment demand a novel efficient and sound treatment [13,14]. The advancement in knowledge about the HCV life cycle and its replication led to the development of HCV protease inhibitors [15,16].

A polyprotein precursor encoded by HCV RNA genome contains structural proteins capsid [C], membrane [prM], envelope [E], and nonstructural (NS) proteins (NS1, NS2a, NS2b, NS3, NS4a, NS4b, NS5) [17] (Figure 1). 

NS3 protease was found to be very supportive in the replication of the HCV virus as it causes the polyprotein to produce the non-structural proteins 4A, 4B, 5A, and 5B when activated by NS4A [18,19]. Therefore NS3/4A protease is considered to be an important emerging target for the treatment of HCV [20]. NS3 protein of the HCV polyprotein in its full length contains amino acids ranging from 1027 to 1657 [21]. NS3 protease consists of two domains, an N-terminal protease domain and a C-terminal helicase domain [19]. The N terminal domain consists of 180 amino acids ranging from 1027 to 1206 showing the protease activity and the remaining 450 amino acids, that is, from 1207 to 1657 are associated with helicase activity [22,23,24] as shown in Figure 1. In addition to the individual functions of these two domains, it has also been found that protease upturns the helicase activity, and helicase upturns the protease activity [25,26]. The active site of NS3 protease includes the residues His-57 (His-1083), Asp-81 (Asp-1107), and Ser-139 (Ser-1165) [26] as indicated by “ᵜ” in Figure 1. 

Currently, it was found that NS3/4A inhibitors have shown hopeful results in curing HCV-infected patients [27,28]. A number of HCV NS3/4A inhibitors that are passing through clinical improvement have very good potency against HCV infection [29]. However, it was observed that rapid viral resistance is developed in patients treated with such inhibitors [30]. Therefore, to identify and develop more potent inhibitors of HCV NS3/4A protease, continuous efforts are desirable. The Protein Data Bank (PDB) is now presented with a number of free and complex structures of NS3/4A protease [31,32] that provide a very good source for the development of novel and potent therapy against HCV. 

A molecular docking study was performed by docking a subset of seven ongoing phase 1 inhibitors reported from BMS, Enanta/Abbott, BI, and Achillion with the crystal structure of HCV NS3/4A protease (PDB code: 4a92) to better understand the interactions between enzyme binding sites with these inhibitors. Then, two applications “RECAP Analysis and RECAP Synthesis” implemented in MOE (molecular operating environment) [33] were applied to the above subset of seven reported inhibitors as part of a de novo discovery methodology. The purpose of RECAP Analysis is to fragment every molecule in the subset according to simple retrosynthetic analysis rules and collect statistics on the resulting fragments. The RECAP Synthesis application was then used to randomly recombine the RECAP Analysis fragments in an effort to generate synthetically reasonable novel chemical structures. The molecular docking study of these novel chemical structures was then performed, and finally, 12 different compounds were identified as inhibitors of protease and helicase activity of NS3/4A protease.

## 2. Results and Discussion

The computational workflow mentioned in Figure 1 was implemented in order to generate synthetically reasonable novel chemical structures that may act as inhibitors of NS3/4A protease. In the following sections, the computational workflow is described in detail. 

### 2.1. Molecular Docking on Prepared Data Set Inhibitors

Molecular docking was conducted on all the seven ongoing phase 1 inhibitors of HCV NS3/4A protease reported from BMS, Enanta/Abbott, BI, and Achillion, using the molecular docking algorithm implemented in MOE. A good correlation was found between the docking scores and the experimental findings. The molecular docking results are described below (Figure 1).

### 2.2. Analysis of NS3/4A Protease Active Site

As discussed in introductory section that the active site of NS3 protease includes the residues His-57 (His-1083), Asp-81 (Asp-1107), and Ser-139 (Ser-1165) and requires NS4A a hydrophobic peptide for its activation. In transfected cells on the endoplasmic reticulum (ER), a stable complex of NS3 and NS4A, is formed [19] by the interaction of N terminal thirty amino acids of NS3 with NS4A [19,34], as presented in Figure 1. In the complex, NS4A plays a double role by enhancing the proteolytic activity of NS3 and connecting the NS3 protein to the ER membrane [35]. 

Earlier research shows that, structurally, the NS3 protease is similar to the trypsin with the exception of a deep active site, a non-catalytic zinc ion, and its dependence on a viral cofactor NS4A making it unique [17]. A zinc-containing binding site in the NS3 protease was predicted by homology modeling for the first time and was confirmed later on by biochemical analysis [36]. The NS3 protease for its activity depends on the zinc ion [37], whereas the addition of EDTA or a cupric ion [38,39] leads to weak inhibition of the proteolytic reaction. It was identified that the zinc ion is coordinated by three cysteine residues, Cys 97, 99, 145, and a His 149 is located opposite to the active site as shown in Figure 2. Thus, in the development of novel and potent inhibitors of NS3/4A protease, such features forced a huge challenge. Currently, NS3/4A inhibitors were found to be very hopeful contestants for curing HCV infection [23].

### 2.3. Test of the MOE-Dock Algorithm

Before docking the prepared data set inhibitors, the ligand from the complex crystal structure NS3/4A protease (PDB ID: 4a92) was removed and re-docked into the binding cavity of protein to validate the docking protocol. By using the SVL script of MOE the root-mean-square deviation (RMSD) between the co-crystallized and re-docked conformation was calculated as 1.83 Å (Figure 3). The RMSD value, 1.83 Å suggests that our docking protocol is reliable in reproducing the experimentally determined binding mode (interactions, conformation, and orientation) for the corresponding protein–ligand complex. Thus, the MOE docking protocol and the parameters set could be used to explore the binding modes of other compounds accordingly. Using the same docking protocol, all the data set inhibitors were docked into the binding pocket of HCV NS3/4A protease.

### 2.4. Docking of the Data Set Inhibitors

After successful validation of the docking protocol mode all the data set inhibitors were docked into the binding pocket of HCV NS3/4A protease. The aim was to examine the binding mode of these reported phase 1 inhibitors of HCV NS3/4A protease and to compare the predicted binding mode for the novel chemical structures generated by the RECAP synthesis application with these reported inhibitors. The attention has been concerted on the characteristic receptor–ligands interactions, binding, and solvation energies. The predicted binding mode for the inhibitor BILN2061 shows such an orientation of the conformation in which the inhibitor interacts with the key residues of the protease site of the enzyme. The various polar and non-polar groups of BILN2061, as shown in Figure 4A, interact with Lys136, Asp81, and His57 residues of the protease active site forming hydrogen acceptor, polar, and pi–pi interactions, respectively. The predicted docking pose of VX-950 among the data set inhibitors as shown in Figure 4B revealed a simultaneous inhibition of the protease and helicase sites of the enzyme. Two pi–H interactions were observed between the non-polar side chain and the aromatic ring of the compound and the protease active site residues His57 and Lys136. Additionally, a hydrogen acceptor bond of one of the carbonyl oxygen of the compound with NH group of Arg123 of the protease site is observed. There are two other carbonyl oxygens of the VX-950 which functionally participate in two hydrogen bonds with the Gln526 and Asp527 of the helicase site of the enzyme. The inhibitor SCH-503034 is well accommodated in the binding pocket of the enzyme and interacts with the residues His57, Ala157, and His528, as presented in Figure 4C. The pi electrons system of His57 is oriented towards the hydrophobic side chain of the compound and interacts through a pi–H bond. Ala157 and His 528 participate in two hydrogen bonds with the NH and carbonyl oxygen of the ligand, respectively. The predicted binding conformation for the inhibitor ITMN-191 shows a well-fit orientation of the compound in the binding pocket of the enzyme, interacting with the key residues of the protease site, that is, His57, Asp81, and the helicase site Met485 (Figure 4D). Two intermolecular hydrogen donor bonds of His57 were observed with C=O and S=O groups of the ligand. Similarly, two intermolecular hydrogen acceptor bonds were noticed between Met485 and the compound. Additionally, a pi–hydrogen bond of the 4-fluoroisoindoline pi electrons of the compound with the S group of Asp81 is also observed. The predicted docking mode of MK-7009 among the data set inhibitors is shown in Figure 4E which demonstrates that the ligand interacts through some healthy intermolecular hydrogen bonds with the key residues of both protease and the helicase sites of the enzyme. The protease active site residues His57, Gly137, and Ala157 established three intermolecular hydrogen acceptor bonds with different oxygen moieties of the compound. Additionally, a hydrogen donor bond of one of the NH groups of the compound with the carbonyl oxygen of Gln526 of the helicase site is observed. A number of significant interactions were observed in the case of the macrocyclic compound with both protease and helicase site residues (Figure 4F). His 57, Gly137, Ser139, Arg155, and Ala 157 from the protease site participated in interactions with the compound. His 57 formed two pi–hydrogen interactions with the compound, whereas Ser139 and Gly137 were found to be involved in hydrogen acceptor interactions with the oxygen of sulfone group of the compound. Arg155 formed a hydrogen donor interaction with the NH group. Ala 157 was observed to be involved in two hydrogen bonds, a donor hydrogen bond with the NH group and a hydrogen acceptor bond with the carbonyl oxygen of the compound. Similarly, His 528 of helicase site residues was also involved in interactions with the oxygen of azepan-2-one moiety of the compound.

This understanding of the in silico mode of interactions of these ongoing phase 1 inhibitors with the receptor will help to understand the key structural characteristics that could assist in designing potent and selective inhibitors of HCV NS3/4A protease. A number of in silico methods have been developed as a useful tool in the identification and design of novel chemical structures with improved characteristics [40,41].

### 2.5. RECAP Analysis

For the present molecular modeling study, seven ongoing phase 1 inhibitors of HCV NS3/4A protease were fragmented using RECAP Analysis. RECAP Analysis is used to fragment a molecule by breaking some bonds which are likely to be those that can be reformed by common reliable chemistry. Thirty-one different fragments were generated from the above-mentioned seven inhibitors according to simple retrosynthetic analysis rules, and statistical data on different fragments was collected. For example, in each RECAP fragment, the number of attachment points to other fragments was calculated. In our generated fragments, the number of attachment points was found to range from 1 to 3 per fragment. Similarly, for each RECAP fragment, the frequency of occurrence was also determined. The frequency of occurrence determined the popularity of a fragment; therefore, the frequency occurrence with descending option was used to determine the most popular fragments. After RECAP Analysis, the results of the analysis databases, that is, fragments in mdb file format were used as the input file to the RECAP Synthesis application.

### 2.6. RECAP Synthesis and Molecular Docking of the Generated Compounds

After RECAP Analysis, the fragments generated were used as an input databases to the RECAP Synthesis application. Using the Molecule Generation Algorithm, the “RECAP Synthesis” randomly recombine the engendered fragments and generate synthetically reasonable novel chemical structures. The RECAP Synthesis was subjected to satisfy a descriptor filter “lead like” with a threshold condition less than 0.5 so that the generated molecules with properties outside the specified range are omitted. Moreover, to know the drug-like ability of newly generated molecules, the properties of molecules were also subjected to Lipinski’s rule of five. Lipinski’s “rule of five” states that drug-like molecules should contain a molecular weight less than 500 Da, a log P value less than 5, hydrogen bond donors less than 5, and hydrogen bond acceptors less than 10 otherwise they have poor permeation or absorption [42]. Following the above-mentioned parameters, the RECAP synthesis application was allowed to generate a maximum of 500 novel chemical structures from 31 different fragments generated in the RECAP Analysis step. These 500 generated novel structures were then prepared according to the procedure as stated earlier in the “Inhibitors Data Set Preparation” section for molecular docking into the active site of the target protein. Using the same docking protocol as used for the data set inhibitors docking, all the 500 newly generated molecules were docked into the binding pocket of HCV NS3/4A protease. A maximum of 10 conformations were allowed to be saved for each ligand using the default parameters of MOE as discussed in the Materials and Methods section. The conformations ranked at the top for all docked compounds were saved in a separate database. Based on the docking score, 200 top-ranked compounds were selected for further evaluation. The predicted intermolecular interactions between these 200 top-ranked compounds and proteins were observed visually using the LigPlot tool implemented in MOE. Those molecules displayed significant interactions with most of the important binding pocket residues (His 57, Val132, Lys 136, Ser 139, Gly 137, Arg 123, 155, Ala 157, Ala 156, Cys159 of protease site and Met 485, Val 524, Glu526, Asp527, and His 528 of helicase site) of HCV NS3/4A protease were selected as promising candidates. Among these 200 molecules, 32 showed vital interactions with the important residues of the target protein. To further refine the durability of these 32 molecules as medicine, these were subjected to binding energy and binding affinity calculations.

### 2.7. Solvation Energy and Binding Affinity Calculations

To identify the ligands with good inhibition potential, for all the 32 shortlisted compounds binding affinities and solvation energies were calculated with generalized born/volume integral (GB/VI) implemented in MOE. Before calculating the binding affinities and solvation energies, an energy minimization of binding pocket in NS3/4A protease–ligand complex was performed in each case. The selection of the most hopeful candidates was based on the criteria that compounds having solvation energies and binding affinities good or equal to that calculated for the dataset inhibitors and showing interactions with key residues in binding cavity of HCV NS3/4A protease. Following the above-mentioned criteria, only 12 of the 32 compounds fulfill the particular requirements (Table 1). The binding mode, binding affinity, solvation energy, and visual prediction showed that these designed lead structures might act as novel, potent, and structurally diverse inhibitors of HCV NS3/4A protease. The 2D structures of these retrieved hits are shown in Figure 5.

### 2.8. Binding Interactions of Finally Selected Compounds

It was found in the docking studies that all final-selected structures showed significant binding interactions with the important residues of protease as well as helicase site of the target protein. For example, compound **1**, for which the strong binding affinity (−8.95 KJ/Mol), lower binding energy (−58.56 KJ/Mol), and good docking score (−9.4824) were observed, displayed the interactions with the protease and helicase binding site residues. The top-ranked docked conformation revealed that the NH group of terminal aliphatic amine and the adjacent alkyl chain of compound **1** interact with the helicase binding site residues of the target protein, whereas the terminal amide group (NH) and carbonyl oxygen atom of carbamate interact with the residues of the protease binding site. The helicase binding site residue, Val524, made an induced polar hydrogen interaction with the N of terminal aliphatic amine, and Met485 made two polar interactions with the same terminal aliphatic amine N and the adjacent alkyl chain. From the protease site, the residues Ser139, Gly137, Ser138, and Val132 are involved in hydrogen bonding with carbonyl oxygen and amide NH_2._ His57 was also observed in a position of H–pi contact as shown in Figure 6A.

The top-ranked docking pose of compound **4** showed that NH group of cyclopropylsulfonyl moiety of the compound showed interactions to Gln526 of helicase site residues of the enzyme (Figure 6B). The protease site residue, Cys159, showed hydrogen bonding with the carbonyl oxygen atom and a polar interaction of terminal sulfonyl (S=O) of the compound. Moreover, Ala156 and Phe438 were also found within a distance of non-polar interaction with at two ends of the compound. In the case of compound **5**, the Gln526 helicase site residue of the enzyme was observed in polar interactions with the compound (Figure 6C). From the protease site, the residues Cys159 and Ala157 are involved in different polar interactions with the compound. Cys159 formed two hydrogen bonds with the two NH_2_ of the carbamoyl group. 

According to the docking study, Compound **6** also showed significant interactions with important residues of protease and helicase sites. For example, in compound **6** it was observed (Figure 6D) that cyclopropylsulfonyl amine and hydroxyethyl amino groups of the compound formed hydrogen bonds to Cys159 and Arg123 of protease site residues, respectively. Similarly, helicase site residues, His 528 and Asp527 were also observed to be involved in hydrogen bonding with cyclopropylsulfonyl and cyclohexyl amino groups of the compound, respectively. 

A number of important interactions were also observed with both protease and helicase site residues in the case of compound **7** (Figure 6E). His 57, Asp81, Arg155, and Cys159 from the protease site were observed to be involved in interactions with the compound. His 57 made an arene–hydrogen interaction with the thiazol moiety of the compound. Asp81 and Arg155 were involved in polar interactions with the thiazol group, whereas Cys159 formed two hydrogen interactions with the NH of the carboxamide group. His528 from the helicase site was found to be involved in hydrogen bonding with the carbonyl oxygen of the carbaxamide group. Similarly, compound **8**, comprising a number of electron-rich species, that is, NH, S, O, also showed promise with both protease and helicase site residues (Figure 6F). Protease site residues, Cys159 and Arg123 presented hydrogen bonding with amino and sulfonyl groups of the compound, respectively. Whereas His528 and Asp527 supported hydrogen bonding from the helicase site. His528 showed two hydrogen bonds with the central diaminohydroxy group, one with OH group and the other with the NH group. Asp527 was found to be involved in hydrogen bonding with the sulfonyl group of the compound. A similar strong hydrogen bonds commitment was also found for compound **12** with the difference that here Ala157 is involved instead of Arg123 and Gln526 instead of Asp527. From the top-ranked docking conformations of all the selected novel structures, it was observed that all the selected structures are able to assume suitable orientations within the binding pocket of HCV NS3/4A protease with some specific functional groups that interact with the important residues of the protease and helicase site of the enzyme. 

## 3. Materials and Methods

As mentioned above, an in silico approach was followed to study HCV NS3/4A protease inhibitors and to predict the inhibitory potential of recently designed chemical structures. The computational process followed is graphically represented in Figure 2.

A data set of the already reported HCV NS3/4A protease inhibitors from BMS (www.bmsmedical.com, accessed on 16 January 2022), Enanta (www.enanta.com, accessed on 16 January 2022), Abbot (www.abbot.com, accessed on 16 January 2022), BI (www.drug.com, accessed on 16 January 2022), Achillion (www.craft.co/achillion-pharmaceuticals, accessed on 16 January 2022), was first docked into the NS3/4A protease active site for calculating the docking score, binding affinity, binding mode, and solvation energy. Then the data set of these reported inhibitors was used in a computer-based program, “RECAP Analyses”, implemented in MOE to fragment every molecule in the subset according to simple retrosynthetic analysis rules. The RECAP Analysis fragments were then used in another computer-based program “RECAP Synthesis” to randomly recombine and generate synthetically reasonable novel chemical structures. The novel chemical structures thus produced were then docked into the active site of HCV NS3/4A protease. After a thorough validation of all undertaken steps, finally, on the basis of Lipinski’s rule of five, docking score, binding affinity, solvation energy, and Van der Waal’s interactions with the protease as well as helicase site residues of target protein **12** novel chemical structures were identified as HCV NS3/4A protease inhibitors.

### 3.1. Inhibitors Data Set Preparation

For the present molecular modeling study, seven ongoing phase 1 inhibitors of HCV NS3/4A protease reported from BMS, Enanta/Abbott, BI, and Achillion were considered. The structures of these inhibitors were built using the MOE molecular builder. All the built structures were 3D protonated and were subsequently energy minimized using the MOE energy minimization parameters, that is, gradient: 0.05 Kcal Mol^−1^ Å^−2^, Force Field: MMFF94X with non-bonded interactions cutoff of 8.0 to 10 Å, using a distance-dependent dielectric function with a dielectric constant of 1. All the compounds were then saved as a mdb file for further evaluation in molecular docking.

### 3.2. Preparation of Protein 3D Structure

The high-resolution X-ray crystal structures of a macrocyclic HCV NS3/4A protease inhibitor complex with their full-length target (PDB ID 4a92) were considered for molecular study. It is familiar that proteins are flexible and can adopt different conformations upon the binding of small molecule inhibitors due to induced fit effects. However, no considerable conformational changes were observed in the crystal structure. The 3D structure of the protein was prepared using the Protein Preparation protocol implemented in MOE 2016. All the hydrogen atoms were added, and all the water molecules were removed. The energy minimization was then carried out for the stability of the protein by using the default parameters of MOE gradient: 0.05 Kcal Mol^−1^ Å^−2^, Force Field: AMBER99. The prepared structure was saved in the PDB file format and was used in molecular docking.

### 3.3. Molecular Docking of Prepared Data Set Inhibitors

MOE’s Dock application 2016 was used for molecular docking procedure. The algorithm that is implemented is arranged as a series of self-contained stages, each of which, if desired, may be bypassed. At most stages, there is a choice from among multiple methods, that is, The GOLD, Surflex, and Flex docking programs. MOE’s Dock application supports favorable binding modes between small to medium-sized ligands and a not-too-flexible macromolecular target, which is usually a protein. For each ligand, a number of placements called poses can be generated and scored. The score can be calculated as either free energy of binding including, among others contributions, solvation and entropy terms or enthalpy terms based on polar interaction energies (including metal ligation) or as a qualitative shaped-based numerical value. The atoms of the active site can be permitted to move: side chains can be tethered, but backbone atoms are always fixed. The final highest scoring poses, along with their scores and conformation energies, are written to a database where they are ready for further analysis. The prepared compounds were docked into the binding site of the target protein in MOE with the parameters, that is, Placement: Triangle Matcher, Rescoring 1: London dG, Refinement: Forcefield, Rescoring 2: GBVI/WSA. There were 10 different conformations generated for each ligand–protein complex. The top-ranked conformation based on the docking score was used for further analysis. The docking score is the binding free energy calculated by the GBVI/WSA scoring function which is the score of the last stage showing the overall fitness of the ligand in the binding pocket. The complex was analyzed for the intermolecular interactions, and their 3D images were drawn by using LigPlot visualizing tool implemented in MOE.

### 3.4. RECAP Analysis

RECAP Analysis is used to fragment a molecule by breaking some bonds which are likely to be those that can be reformed by common reliable chemistry. The methodology is similar to that used by Schneider G, et al. [43] and Weininger D, et al. [44]. A unique extended SMILES [45] name is given to each resulting fragment that retains the chemical context of the broken bond. Using this fragmentation methodology on a set of compounds, statistical data on different fragments can be collected, and the fragments then can be recombined randomly in the RECAP Synthesis to generate novel chemical structures. For the present molecular modeling study, seven ongoing phase 1 inhibitors of HCV NS3/4A protease reported from BMS, Enanta/Abbott, BI, and Achillion were fragmented using RECAP Analysis. Thirty-one different fragments were generated from the above-mentioned seven inhibitors according to simple retrosynthetic analysis rules implemented in MOE. 

### 3.5. RECAP Synthesis and Molecular Docking of the Generated Compounds

After RECAP Analysis the fragments generated were used as an input database to the RECAP Synthesis application. The “RECAP Synthesis” used the Molecule Generation Algorithm to randomly recombine the analysis fragments and generate synthetically reasonable novel chemical structures. The newly generated molecules were used to satisfy a descriptor filter “lead like” with a threshold condition less than 0.5 so that the molecules with properties outside the specified range are omitted. The RECAP synthesis application was allowed to generate a maximum of 500 novel chemical structures from 31 different fragments generated in the RECAP analysis step. These 500 generated novel structures were then prepared according to the above-mentioned procedure (Inhibitors Data Set Preparation) for molecular docking into the active site of the target protein. The docking was performed by the method mentioned above.

### 3.6. Solvation Energy and Binding Affinity Calculations

To identify the most potential leads among the docked conformations of newly generated compounds, solvation energies and binding affinities of the ligands–protein complexes were calculated with the generalized Born/volume integral (GB/VI) implicit solvent method implemented in MOE [46]. The generalized Born interaction energy is the non-bonded interaction energy between the receptor protein and the ligand that includes Van der Waals interactions, Coulomb electrostatic interaction, and implicit solvent interaction energies. Solvent molecules were ignored during calculation. The estimated binding affinity is that of the London dG scoring function reported in units of Kcal/Mol. The atoms of the receptor molecule away from the binding pocket were kept rigid, whereas receptor atoms in the binding site were kept flexible but were subjected to tether restraints that discourage gross movement. The ligand atoms were allowed to be flexible at the binding pocket. The binding affinity was calculated in units of Kcal/Mol for each structure after energy minimization.

## 4. Conclusions

This study was carried out with the aim to model novel chemical structures for the simultaneous inhibition of protease and helicase activities of HCV NS3/4A protease. In this regard, a molecular docking study was conducted by docking a subset of seven ongoing phase 1 inhibitors reported from BMS, Enanta/Abbott, BI, and Achillion with the crystal structure of HCV NS3/4A protease (PDB code: 4a92) to better understand the interactions between enzyme binding site and these inhibitors.

Then, two applications “RECAP Analysis and RECAP Synthesis” implemented in MOE (molecular operating environment) were applied to the above subset of seven reported inhibitors as part of a de novo discovery methodology. The purpose of RECAP Analysis is to fragment every molecule in the subset according to simple retrosynthetic analysis rules and collect statistics on the resulting fragments. The RECAP Synthesis application was then used to randomly recombine the RECAP Analysis fragments in an effort to generate synthetically reasonable novel chemical structures. After a thorough validation of all undertaken steps, finally, based on Lipinski’s rule of five, docking score, binding affinity, solvation energy, and Van der Waal’s interactions with the protease, as well as helicase site residues of the target protein, 12 novel chemical structures were designed as inhibitors for the simultaneous inhibition of HCV NS3/4A protease and helicase site important residues. These novel chemical structures can be used as such and on further optimization as potential leads in developing novel inhibitors of HCV NS3/4A protease. These novel structures comprising unique scaffolds have a strong probability to act as further starting points in the development of novel and potent NS3/4A protease inhibitors.

## Data Availability

The data presented in this study are available on request from the corresponding author.

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
