# Peer review of "A Novel Approach to Develop New and Potent Inhibitors for the Simultaneous Inhibition of Protease and Helicase Activities of HCV NS3/4A Protease: A Computational Approach"

_molecules, 2023, doi:10.3390/molecules28031300_

Round 1

Reviewer 1 Report

the article with title " A Novel Approach to Develop New and Potent Inhibitors for 2 the Simultaneous Inhibition of Protease and Helicase Activities 3 of HCV NS3/4A Protease: A computational Approach" show a good results as dry lab applications and Computational and Theoretical Chemistry, the quality of the figures are good but

1-the author didn’t mention the origin of the tested compound or at least reference for them.

2- the reference should be renewed as I found a reference from 1989 , 2001, 2004   

Author Response

Please see the attached file for comments.

Reviewer 2 Report

Dear Editor,

The study, entitled ‘A Novel Approach to Develop New and Potent Inhibitors for 2 the Simultaneous Inhibition of Protease and Helicase Activities 3 of HCV NS3/4A Protease: A computational Approach’ with manuscript number molecules-2139525, discusses the design of 12 new NS3/4A protease inhibitors via the employment of computer-aided techniques. The study is interesting for researchers in the relevant field. Some minor suggestions are provided below:

- Hep C is a worldwide threatening disease; therefore more statistics should be given in the abstract part regarding the cases in Europe, USA, for instance.

- It is not clear in the abstract part which side of the protein is targeted and how this is rationalized.

- Scheme 2 misses two describe the precursor molecules. Caption should be revised.

- RECAP Analysis is used to fragment a molecule by breaking some bonds which are 169 likely to be those that can be reformed by common reliable chemistry. The methodology 170 is similar to that of [34] and [35]. This sentence does not make sense. 34 and 35 should be clarified in the text.

- Line 177 Thirty one different fragments 177 were generated from the above mentioned seven inhibitors according to simple retrosynthetic analysis rules. Citation is missing.

-   Figure 4 also misses sufficient caption to introduce the compounds.

- What is the importance of selected drug-likely properties?

- I don’t think there is need for the nomenclature of the 12 title compounds.

- Although it is a computational study, one wonders at least one of those compounds exact activity measured.
